# Adsorption of oxytetracycline on kaolinite

Yali Song[1,2], Ebenezer Ampofo Sackey[1], He Wang[1], Hua Wang[1,2]*

**1** School of Civil Engineering and Architecture, Zhejiang University of Science and Technology, Hangzhou, Zhejiang, China, **2** Key Laboratory of Recycling and Eco-treatment of Waste Biomass of Zhejiang Province, Zhejiang University of Science and Technology, Hangzhou, Zhejiang, China

\* wangh08@hotmail.com

**Data Availability Statement:** All relevant data are within the paper and its Supporting Information files.

**Funding:** This study was financially supported by the Zhejiang Provincial Natural Science Foundation of China (No. LY16E080007) to YS and the Public

## Abstract

As antibiotic contamination increases in wastewater and aqueous environments, the reduction of antibiotics has become a pertinent topic of research regarding water treatment. Clay minerals, such as smectite or kaolinite, are important adsorbents used in water treatment, and sufficient removal of antibiotics by clay minerals is expected. In this study, the adsorption of oxytetracycline (OTC) on kaolinite was investigated. The experimental data of OTC adsorption on kaolinite fit the pseudo-second-order kinetics model well ($R^2$>0.98). After 24 h, adsorption equilibrium of OTC on kaolinite was reached. The Langmuir model was better fitting with the adsorption isotherms generated from experimental data and OTC adsorption occurred on the external surface of kaolinite. The analysis of several thermodynamic parameters indicated that the adsorption of OTC on kaolinite was spontaneous and thermodynamically favorable. With the increase of the pH of a solution, the adsorption capacity increased and then decreased. The adsorption coefficient ($K_d$) of $10^2$–$10^3$ were obtained for adsorption process of OTC on kaolinite.

## Introduction

Currently, antibiotic contamination in the environment has received considerable attention [1–3]. Many antibiotics have been detected in wastewater and surface water [4–6], which results in the deterioration of the aquatic environment and the production of antibiotic resistant bacteria [7, 8]. Oxytetracycline (OTC) is a member of tetracyclines (TCs) antibiotics which are used widely in the world. The presence of OTC in wastewater treatment plant effluents, aqueous environments and even in drinking water has been reported in some studies [9–11]. Therefore, it is important to develop an efficient method to remove OTC from the aqueous phase. Several studies to remove OTC during water treatment have been reported [12–14].

Adsorption is widely used for pollutant removal during water treatment processes. Some studies have reported the adsorption of TCs by adsorbents such as activated carbon, biochar and clay minerals [15–17]. Previous studies on interactions between TCs and clay minerals have mostly focused on smectite because of a high cation exchange capacity and big surface area of smectite [18, 19]. However, the adsorption capability of TCs on kaolinite has less literature reports owing to its low cation exchange capacity and surface area. Figueroa et al reported that the adsorption capacity of kaolinite was lower than that of montmorillonite [20]. Some

Welfare Technology Application Research Project of Zhejiang Province (No. 2016C33102) to HW.

**Competing interests:** The authors have declared that no competing interests exist.

studies of the interaction of antibiotics and kaolinite mainly involved factors such as pH, organic matter or ionic strength. Zhao et al investigated the effects of some factors such as pH, background electrolyte cations and humic acid on kaolinite for TC adsorption, and the results indicated that TC adsorption by kaolinite was influenced by changes in the above mentioned solution conditions [21].The study on the interactions between TC and kaolinite found that the TC adsorption on kaolinite mainly focused on cation exchange of the external surfaces rather than due to complexation [22]. Some studies on kaolinite adsorbing TC have been carried out; however, few studies on the adsorption of OTC on kaolinite have been conducted. In this study, the adsorption capacities of OTC on kaolinite were investigated, and the adsorption mechanism was discussed for OTC on kaolinite.

## Materials and methods

### Materials

In this study, oxytetracycline hydrochloride was purchased from Dr. Ehrenstorfer (Germany). Acetonitrile (HPLC grade) and methyl alcohol (HPLC grade) were purchased from Merck (America). The kaolinite sample was obtained from Macklin (Shanghai). The kaolinite samples were filtered through a 300 mesh sieve and no further purification. The specific surface of kaolinite sample was 4.3 $m^2$/g measured with the $N_2$/BET method by ASAP 2020 Plus of Micromeritics (America). The measure of the particle size was conducted with an automatic laser particle size analyzer (LAP-W2000H, Xiamen), and most of the kaolinite was approximately 2.1 μm in size. The infrared spectroscopic analysis was conducted by fourier transform infrared spectrometer (VERTEX 70, Bruker, Germany) in the 400–4000 $cm^{-1}$ wavenumber range. KBr pressed-disc method was adopted in this study. Kaolinite treated with OTC was pretreated with freeze drying. A certain mass of dried sample and KBr were mixed and grinded together in an agate bowl. Then, the mixed powder was pressed into discs and detected with spectrometer.

### Adsorption experiments

The kinetic adsorption of OTC on kaolinite was conducted in a batch experiment. Accordingly, 40 mg kaolinite and 20mL 0.01 M $CaCl_2$ electrolyte solution with different OTC concentration (at 5, 10 and 25 mg/L) were combined in 40 mL brown glass vials, and the mixed samples were shaken on a reciprocal shaker at 150 rpm and 298 K for 0.16, 0.33, 0.3, 1, 2, 4, 6, 8, 12, 16, 20, 24, 36 and 48 h at pH 5.5. The above samples were centrifuged at 5000 rpm for 10 min, and the supernatant solution was filtered through a 0.22μm membrane. Subsequently, the OTC concentration of the filtered solution was determined by HPLC with a UV detector (e2695, Waters, USA).

The initial OTC concentrations of 1 to 35 mg/L were used to generate adsorption isotherms under three temperature conditions (288 K, 298 K and 308 K). Then, 20 mL of OTC solutions at different concentrations and 40 mg of kaolinite were mixed at pH 5.5 in 40 mL brown glass vials. The mixed samples were shaken with the shaker at 150 rpm until reaching adsorption equilibrium. The OTC concentration in the equilibrium samples was obtained using the same method as described above (refer to kinetic adsorption).

The influence of solution pH on the OTC adsorption by kaolinite was evaluated in a series of batch experiments. The variation of solution pH was from 3.5 to 11.5, and the concentration of OTC was 5 mg/L or 10 mg/L. pH of OTC and kaolinite mixed samples were adjusted according to the above series of pH values and then adjusted samples were shaken at 150 rpm to adsorption equilibrium. Finally, the OTC concentration was measured.

**Table 1. Adsorption kinetics and isotherms models.**

| Adsorption model | | Model equation | Parameters |
|---|---|---|---|
| **kinetic models** | pseudo-first-order | $\log(q_e - q_t) = \log q_e - \frac{k_1 t}{2.303}$ | $q_e$: the adsorption capacity at the equilibrium time (mg/g)<br>$q_t$: the adsorption capacities at the time t (mg/g)<br>$k_1$: the rate constant of the pseudo-first-order model (h$^{-1}$) |
| | pseudo-second-order | $\frac{t}{q_t} = \frac{1}{k_2 q_e^2} + \frac{t}{q_e}$ | $k_2$: the rate constant of the pseudo-second-order (g/(mg·h)) |
| **isotherm models** | Langmuir | $q_e = \frac{k_l q_{max} C_e}{1 + k_l C_e}$ | $C_e$: the OTC equilibrium concentration (mg/L)<br>$k_l$: the adsorption coefficient<br>$q_{max}$: the maximum adsorption capacity(mg/g) |
| | Freundlich | $q_e = k_f C_e^n$ | $k_f$: the sorption coefficients (mg$^{1-n}$·L$^n$/g)<br>n: the linear coefficients |
| | Tempkin | $q_e = \frac{RT}{B_T} \ln(k_t C_e)$ | $k_t$: the Tempkin constant that corresponds to the maximum binding energy (L/mg)<br>$R$: universal gas constant (8.314 J/(mol·K))<br>$T$: the absolute temperature (K)<br>$B_T$: obtained after solving the Tempkin equation |

Control samples including no kaolinite or no OTC conditions were conducted simultaneously in each experiment and all experiments were run in triplicate.

## Data analysis

Adsorption kinetics and isotherms models are showed in Table 1.

Thermodynamic analysis was conducted using Eqs (1)–(3) to obtain thermodynamic parameters, such as the standard Gibbs free energy ($\Delta G$), enthalpy ($\Delta H$) and entropy ($\Delta S$), as follows:

$$\Delta G = -RT \ln K_d \qquad (1)$$

$$\Delta G = \Delta H - T\Delta S \qquad (2)$$

$$\ln K_d = -\Delta H/RT + \Delta S/R \qquad (3)$$

Where $T$ (K) is the absolute temperature and $R$ (8.314 J/(mol·K)) is the universal gas constant.

## Results and discussion

### OTC adsorption kinetics

The adsorption kinetics of OTC by kaolinite is shown in Fig 1. For OTC, the adsorption capacity of kaolinite had a varying trend with increasing time. During the initial adsorption time, apparent adsorption was observed, and the adsorption capacity reached approximately 50% at the sorption time of 4 h. From the figure, a steep curve was presented before 8 h. A previous study found that a substantial mass transfer driving force occurred between the adsorbent and solution because of the high concentration of antibiotic, which caused the antibiotic to rapidly occupy the adsorption sites of the adsorbent [18, 22]. In the range of 8-24h, adsorption capacity increased slowly than that during the initial time with the increment of time. Subsequently, the adsorption capacity changed slightly, and a steady curve could be found during 24–48 h, which indicated that adsorption equilibrium was obtained after 24 h. The adsorption kinetics of the three concentrations of OTC presented the same adsorption trend, and low concentrations of OTC had faster equilibrium times than those of high concentrations. Compared to smectite, the adsorption capacity of OTC on kaolinite is much lower [18]. In this study, OTC

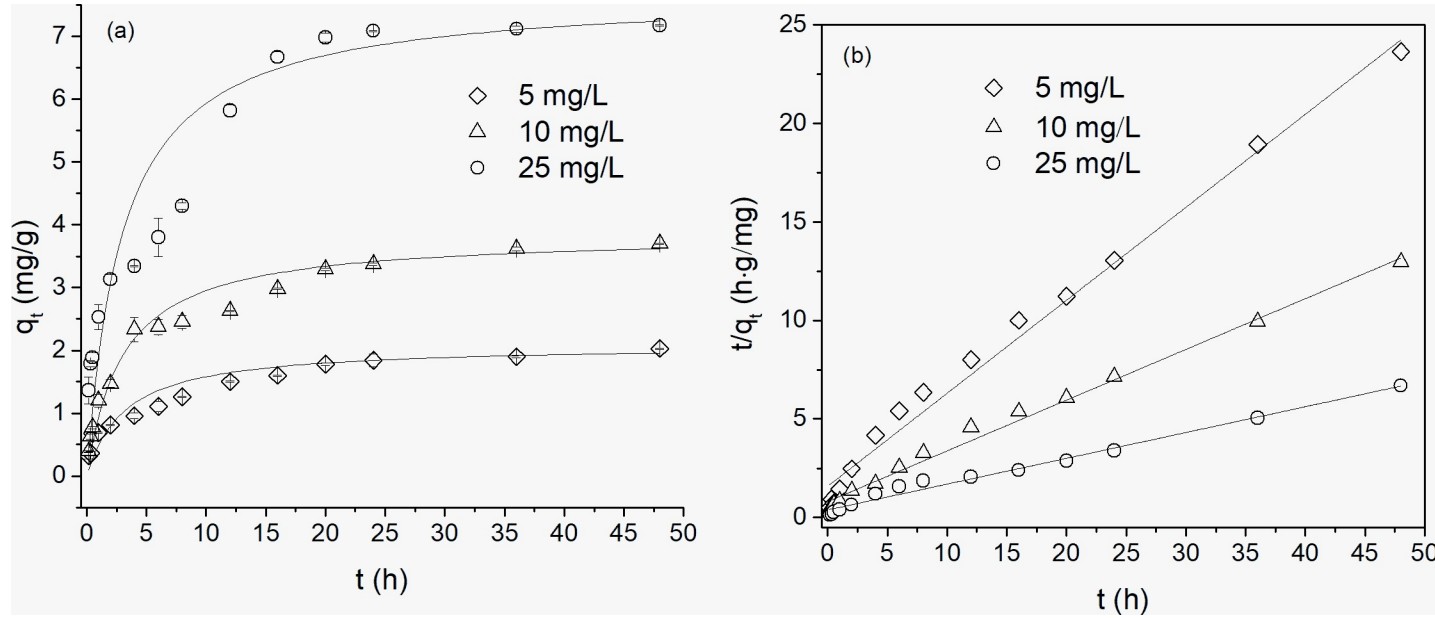

**Fig 1.** Adsorption kinetics of OTC on kaolinite: (a) pseudo-second-order model; (b) the linear plot of the pseudo-second-order model.

adsorption capacity of 7mg/g on kaolinite was obtained during initial OTC concentration of 25mg/L, which is similar to the results of Zhao's study [21].

Table 2 shows the adsorption kinetics parameters of the pseudo-first-order and pseudo-second-order models. The $R^2$ of the pseudo-second-order model of the experimental data reached above 0.98 but was only 0.747–0.873 for the pseudo-first-order model. The experimental data obviously fit a pseudo-second-order model, and the adsorption of OTC on kaolinite mainly involved chemical adsorption processes. This is consistent with previous studies [23]. From the table, the value of $q_e$ of the pseudo-second-order model increased with increasing initial OTC concentration. Moreover, the fitted rate constants of 0.044–0.149 g/(mg·h) during initial OTC concentration of 5-25mg/L were higher than that of TC on smectite. However, compared with TC adsorption on smectite [22], the initial rates of 0.65–2.60mg/(mg·h) in this experiment were much lower.

## Adsorption isotherms

In this study, three models including Langmuir model, Freundlich model and Tempkin model were adopted to evaluate the adsorption equilibrium isotherms of OTC on kaolinite. Fig 2 shows the adsorption isotherms for the adsorption of OTC on kaolinite. With increasing temperature, the capacity of OTC adsorption on kaolinite presented a significant difference: the higher the temperature, the larger the adsorption capacity. The largest adsorption capacity

**Table 2. Pseudo-first-order and pseudo-second-order parameters of OTC adsorption on kaolinite.**

| OTC concentrations | Pseudo-first-order model | | | Pseudo-second-order model | | |
|---|---|---|---|---|---|---|
| | $q_e$ (mg/g) | $k_1$ (h$^{-1}$) | $R^2$ | $q_e$ (mg/g) | $k_2$ (g/(mg·h)) | $R^2$ |
| 5 mg/L | 1.331 | 0.047 | 0.782 | 2.088 | 0.149 | 0.988 |
| 10 mg/L | 0.739 | 0.051 | 0.747 | 3.846 | 0.086 | 0.993 |
| 25 mg/L | 0.386 | 0.059 | 0.873 | 7.692 | 0.044 | 0.986 |

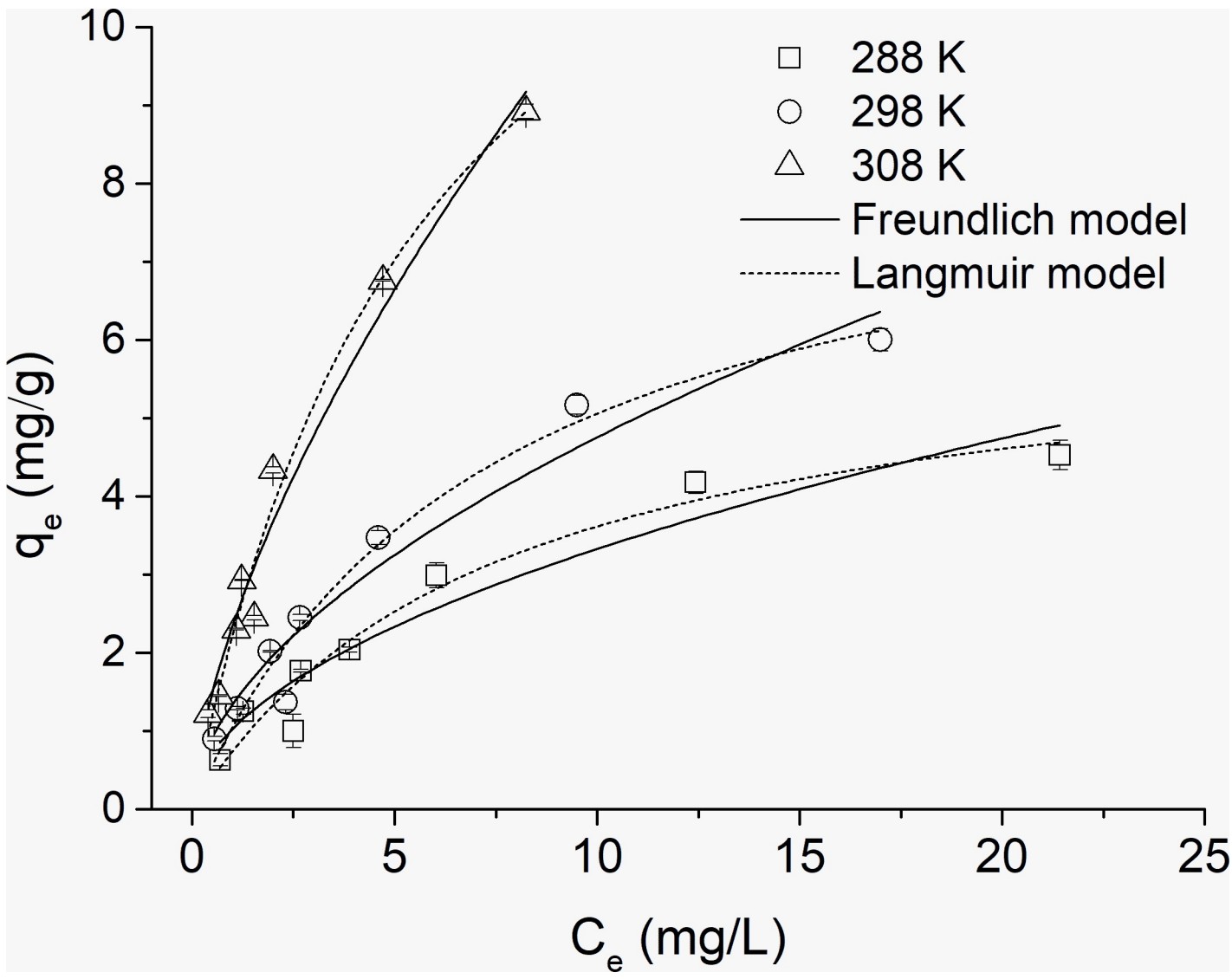

**Fig 2. Adsorption isotherms for the OTC adsorption on kaolinite.**

reached 8.92 mg/g under 308 K condition, and a capacity of only 4.53 mg/g was reached at 288 K.

The experimental data was better fitted to the Langmuir model than Freundlich model, and the correlation coefficient ($R^2$) was above 0.95 (0.953–0.980) for Langmuir model (Table 3). However, the correlation coefficient of the Tempkin model was worse than that of the above two models. This indicated the sorption of OTC on kaolinite was monolayer adsorption and may occur on the external surface of kaolinite [22]. The trend of $q_{max}$ in the Langmuir model is increasing with the increase of temperature. The value of $q_{max}$ changed from 6.342 mg/g to 15.236 mg/g over the range of 288 K-308 K. The adsorption capacity of OTC on kaolinite was similar to that of TC adsorption on kaolinite [22] and quinolone antibiotic nalidixic acid onto kaolinite [24]. The Freundlich model constant n represents the sorption intensity, and an n value <1 implied good adsorption and a concentration-dependent process. In this study, n values <1 were obtained in the Freundlich model.

Table 3. Langmuir, Freundlich and Tempkin models parameters for OTC adsorption on kaolinite.

| Temperature(K) | Langmuir model | | | Freundlich model | | | Tempkin model | | |
|---|---|---|---|---|---|---|---|---|---|
| | $q_{max}$ (mg/g) | $K_L$ (L/mg) | $R^2$ | $K_f$ (mg$^{1-n}$·L$^n$/g) | n | $R^2$ | $K_T$ (L/mg) | $B_T$ (×10$^3$) | $R^2$ |
| 288 | 6.342 | 0.133 | 0.953 | 0.875 | 0.511 | 0.921 | 1.737 | 1.2329 | 0.911 |
| 298 | 8.749 | 0.137 | 0.965 | 1.227 | 0.549 | 0.944 | 1.9513 | 1.6258 | 0.915 |
| 308 | 15.236 | 0.171 | 0.980 | 2.220 | 0.645 | 0.971 | 2.634 | 2.6526 | 0.921 |

To further analyze the mechanisms of adsorption, FTIR spectroscopic analysis was conducted. Fig 3 shows the changes of FTIR spectrum plot for raw kaolinite and kaolinite treated with 10mg/L OTC. The FTIR spectrum of OTC adsorbed by kaolinite changed for the two

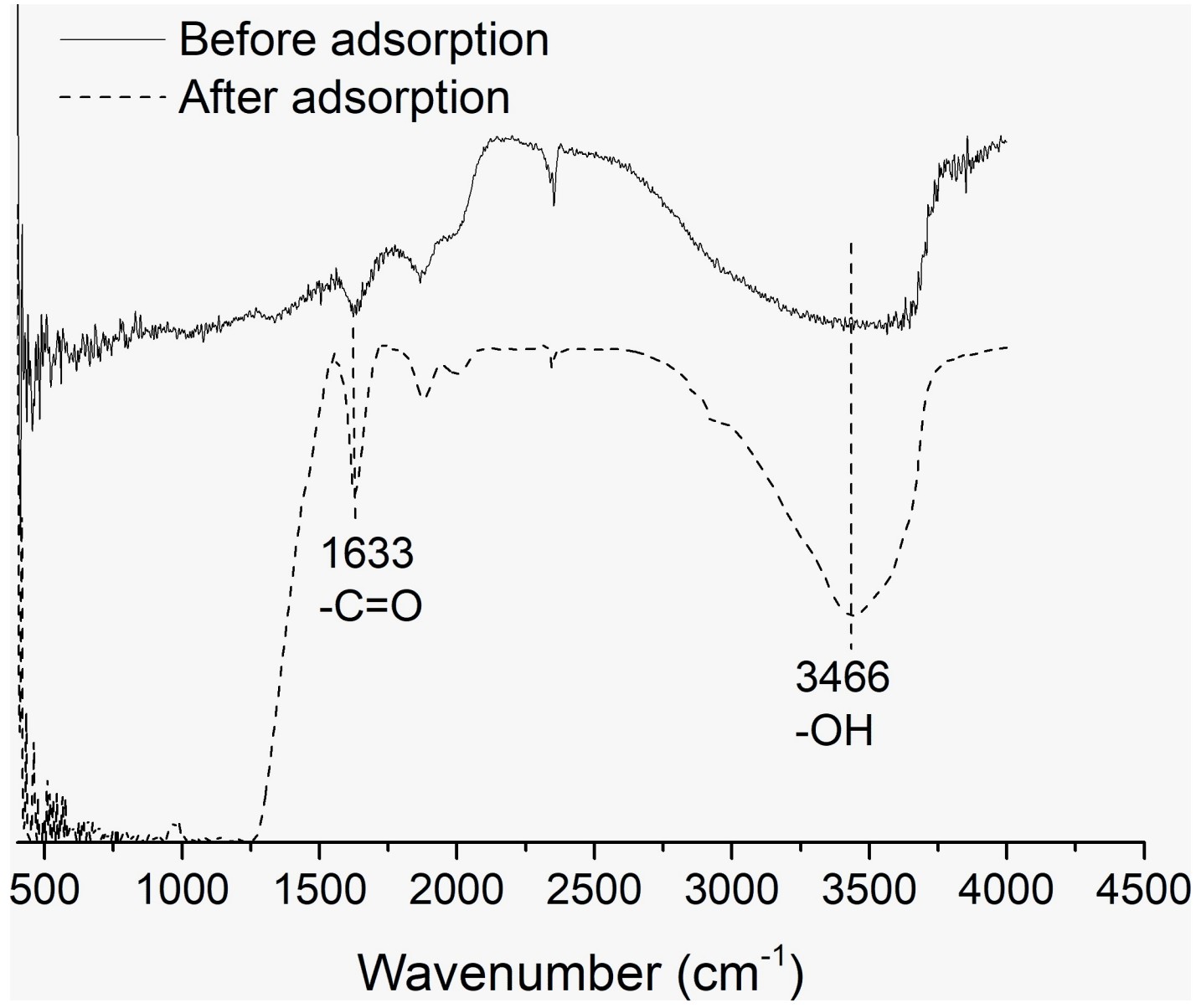

Fig 3. Fourier transform infrared spectroscopic analysis of kaolinite before and after adsorption.

bands. The peak intensity at 1633, corresponding to the–C = O groups, was observed after adsorption which implied that OTC was adsorbed on kaolinite [25]. Jia suggested that the intense stretching band between 3300 and 3500 cm$^{-1}$ is due to the O-H of OTC [26]. In this study, the peak intensity at 3466 increased after adsorbing OTC which indicated the interaction between OTC and kaolinite.

### Analysis of thermodynamics

During the thermodynamics analysis, three thermodynamic parameters including ΔG, ΔH and ΔS were investigated. The effect of temperature on the adsorption coefficient (K$_d$) for OTC sorption on kaolinite is shown in Fig 4. Here, K$_d$ was calculated by q$_e$/C$_e$ (L/kg) and the ΔG values were derived from lnK$_d$. Table 4 shows the parameters of thermodynamics of OTC adsorption on kaolinite. All ΔG values were negative during different adsorption temperatures, which indicated that the adsorption of OTC on kaolinite was spontaneous and thermodynamically favorable. Moreover, the absolute ΔG values increased with the adsorption temperature, which implied that high temperature is favorable for adsorption of OTC on kaolinite. The positive ΔH values of OTC adsorption on kaolinite implied an endothermic adsorption process which agrees with the results of the favored high temperatures indicated by ΔG. The positive ΔS suggested that the adsorption process favored sorption stability. In this study, the ΔS value was 245.64 J/(mol·K), which implied that disorder increased at the interface between OTC and kaolinite during the adsorption process.

### The effect of pH on OTC adsorption

To further analyze the effects of different OTC fractions on the adsorption process, an empirical model (eqn.4) of $K_d$ was used in this study.

$$K_\mathrm{d} = K_d^{+00} \times f^{+00} + K_d^{+-0} \times f^{+-0} + K_d^{+--} \times f^{+--} + K_d^{0--} \times f^{0--} \tag{4}$$

Where $K_d$ is the adsorption coefficient (L/kg); $K_d^{+00}$, $K_d^{+-0}$, $K_d^{+--}$, and $K_d^{0--}$ are the adsorption coefficients of the four OTC fractions; $f^{+00}$, $f^{+-0}$, $f^{+--}$, and $f^{0--}$ are the cationic fraction, zwitterionic fraction, amination anionic fraction and bivalent anionic fraction, respectively.

The effect of pH on the adsorption capacity for OTC sorption on kaolinite is shown in Fig 5. There were substantial influences of pH on the adsorption capacities of kaolinite for OTC. With the increase of pH, the adsorption capacity increased and then decreased. A maximum adsorption capacity existed at a pH of approximately 5.5. The surface charges of kaolinite and OTC changed with different pH values, which influenced the adsorption properties. For amphoteric OTC, there are three pK$_a$ values (3.57, 7.49 and 9.88), and OTC was divided into four fractions as follows under different pH conditions: the cationic OTC (OTC+00) fraction with pH<3.57, the zwitterionic OTC fraction (OTC+-0) during pH of 3.57–7.49, the amination anionic OTC (OTC+--) or bivalent anionic OTC (OTC0--) with pH >7.49 (Fig 6). It is believed that surface of kaolinite had a constant structural charge and edge charge depending on the solution pH [27]. The surface charge of kaolinite is normally considered to be a negative surface charge and some positive charge exists on kaolinite under acidic conditions, while negative charges presents under alkaline conditions [21]. In this study, when the pH was greater than 7.49, the same charges existed on both OTC and kaolinite and resulted in electrostatic repulsion. Thus, the adsorption capacity of kaolinite for OTC in the pH range >7.49 was worse than that during in pH range <7.49. At pH<3.57, the positive charge of OTC can be adsorbed by the negative charge of kaolinite; with the increase of pH value, the charge of OTC becomes neutral. The adsorption capacities presented an increasing trend, which is similar to the results of previous studies [28, 29]. Some studies have suggested that the mechanism is

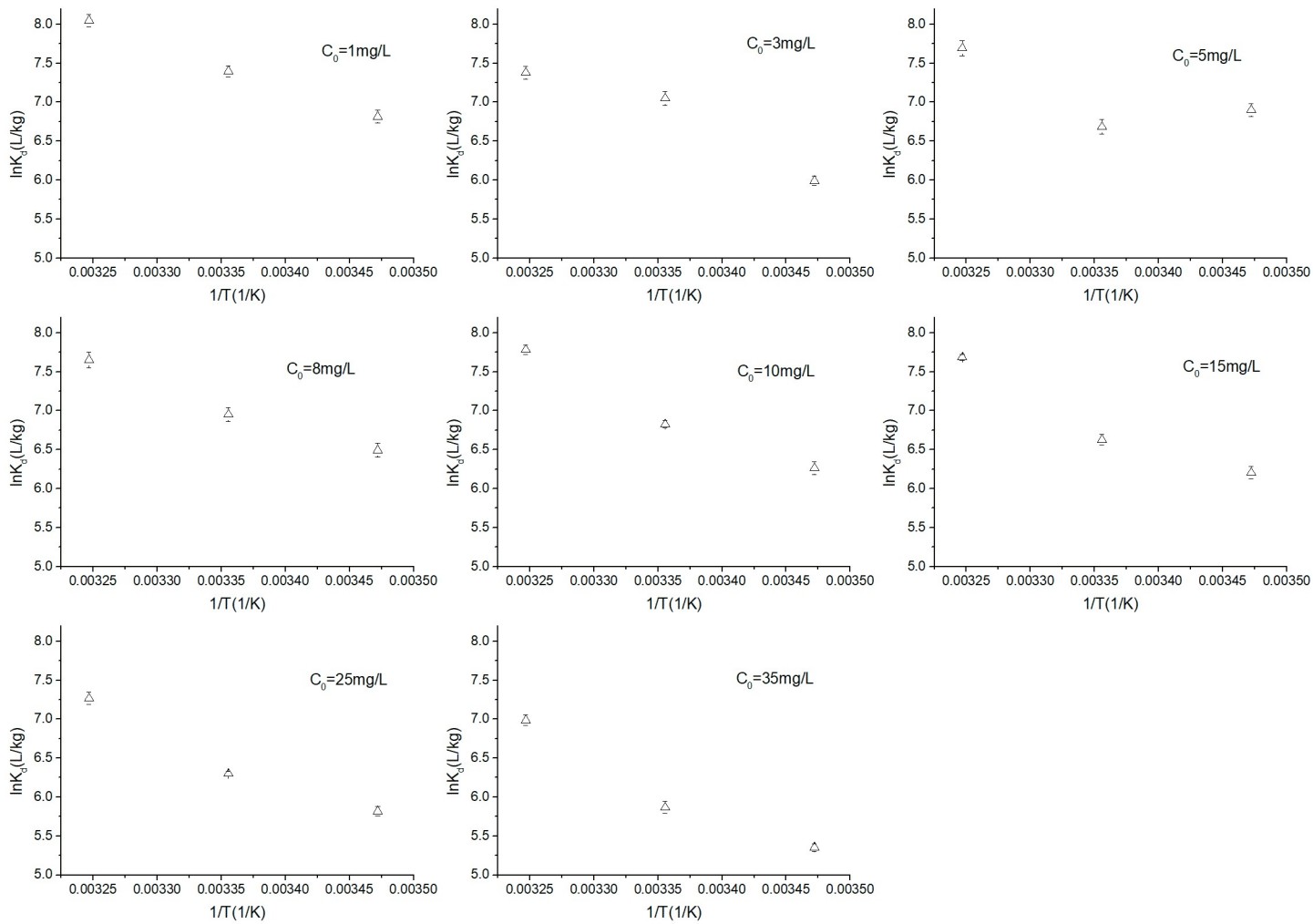

**Fig 4. Effect of temperature on the sorption coefficient ($K_d$) for OTC sorption on kaolinite.**

complexation [20, 30]. However, cation exchange mechanism for antibiotics adsorption on clays was supported by more researchers. Li et al found that there existed simultaneous $H^+$ uptake during the adsorption process of tetracycline onto smectites and cation exchange occurred even under neutral pH conditions [18]. Zhao et al suggested that the adsorption mechanism between tetracycline and the kaolinite surface was similar to an outer-sphere cation exchange reaction [21]. Many studies agreed low pH was beneficial to the adsorption of TC and the adsorption mechanism was cation exchange [20, 22, 29].

The adsorption coefficients ($K_d$) for the OTC species at different pH values are shown in Table 5. A relatively good fit of the $K_d$ data was obtained, and $R^2$ reached 0.826 for 5 mg/L and

**Table 4. Thermodynamic parameters for OTC adsorption on kaolinite.**

| Temperature (K) | ΔH (kJ/mol) | ΔS (J/(mol·K)) | ΔG (kJ/mol) | $R^2$ |
|---|---|---|---|---|
| 288 | 55.92 | 245.64 | -14.99 | 0.972 |
| 298 | | | -16.34 | |
| 308 | | | -18.64 | |

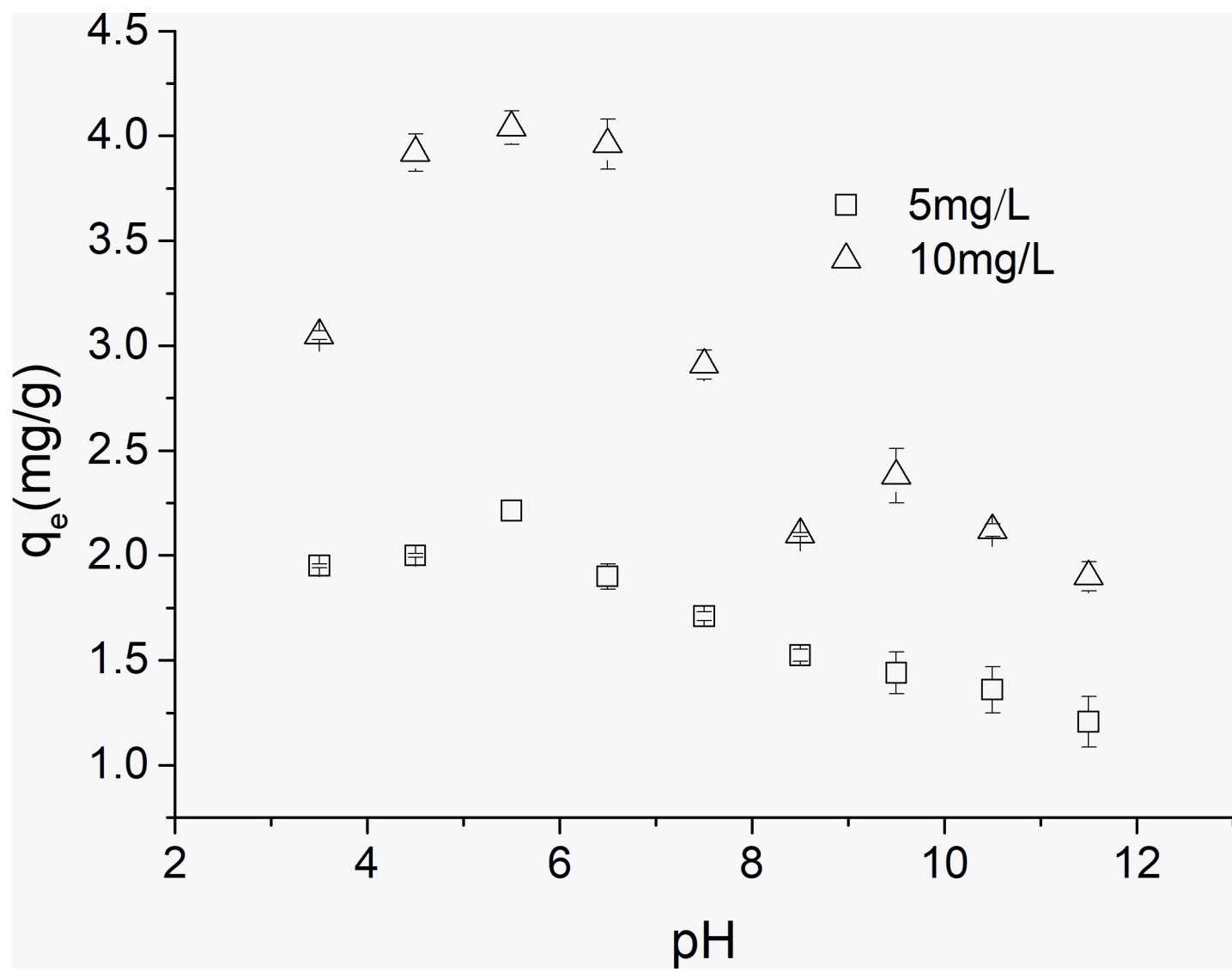

**Fig 5. Effect of pH on OTC adsorption capacity on kaolinite.**

0.982 for 10 mg/L. From the table, it can be seen that the four OTC species exhibited different adsorption coefficients. The value of $K_d^{+-0}$ was greater than the other three values, $K_d^{+00}$ and $K_d^{0-}$ were in the middle, and $K_d^{+-}$ was the lowest. This illustrated that the highest adsorption affinity was obtained by the zwitterionic species during the four species. The contribution results of different species to OTC adsorption indicated that more than 55% contribution rates of the zwitterionic species were obtained for two OTC concentrations (5 mg/L and 10 mg/L). It seems that interaction between the zwitterionic species and the negative surface charge of kaolinite is easy. In addition, positive OTC species had more contribution to OTC adsorption than negative species did, which is also confirmed by the above data (refer to Fig 5).

### Adsorption affinity

Fig 7 shows the adsorption coefficient ($K_d$) values for the OTC adsorption on kaolinite. With an increasing OTC equilibrium concentration, the values of $K_d$ decreased. Obviously, the

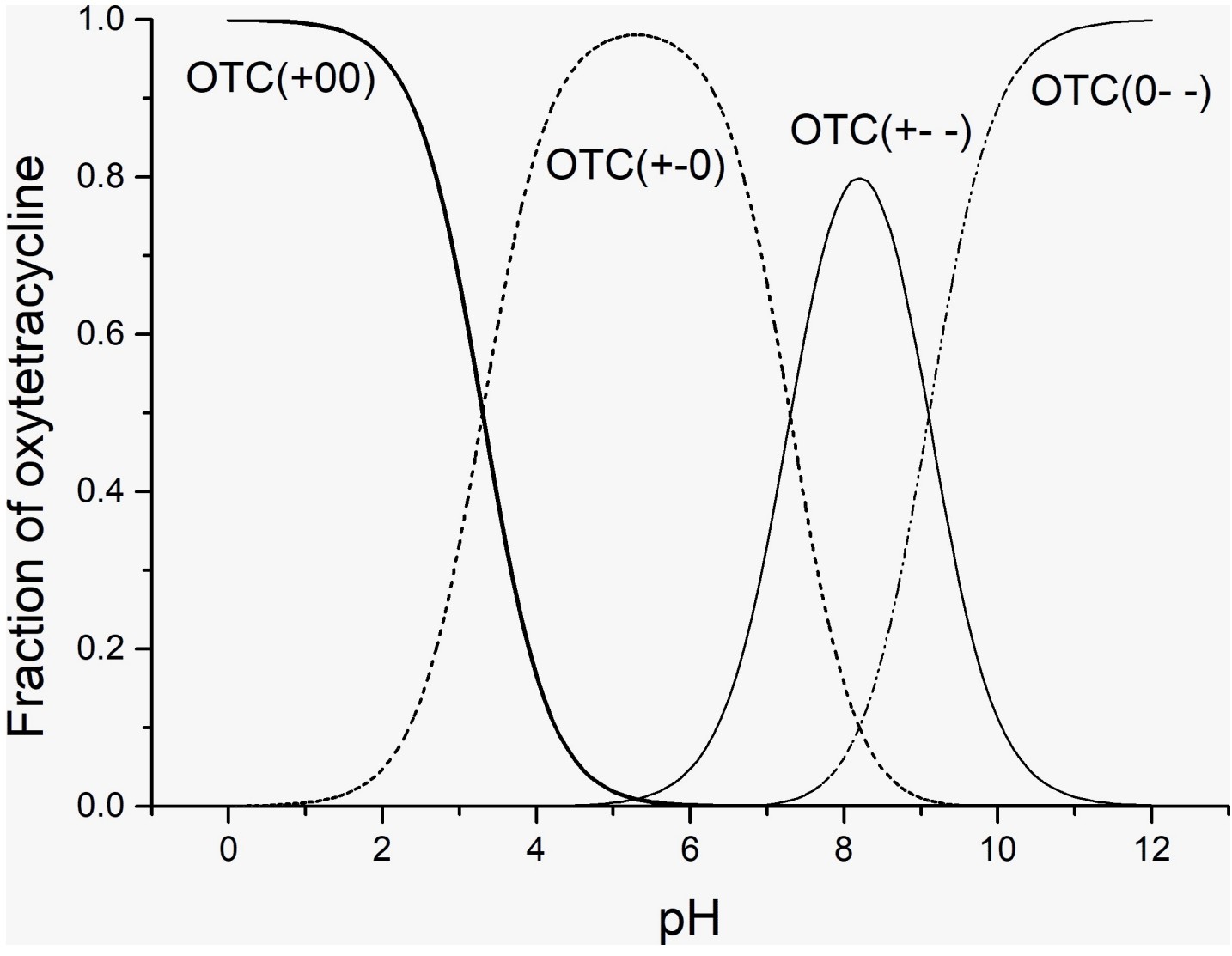

**Fig 6. Distribution of OTC species during different pH values.**

adsorption affinity between OTC and kaolinite was highly correlated with OTC concentration, and the lower the concentration, the larger the adsorption affinity. In this study, the adsorption capacity was evaluated by $K_d$, and the large value of $K_d$ is favorable for adsorption. The study indicated that the $K_d$ values of some natural adsorbents were $10^2$–$10^3$ L/kg [20]. Similar results were obtained in this study, which were $K_d$ values ranging from 350 to 1600 L/kg.

**Table 5. Calculated adsorption coefficients for the OTC species.**

| | $K_d^{+00}$ (L/kg) | $K_d^{+-0}$ (L/kg) | $K_d^{+--}$ (L/kg) | $K_d^{0--}$ (L/kg) | $R_{adj}^2$ |
|---|---|---|---|---|---|
| **5 mg/L OTC** | 976.69 | 2636.21 | 373.59 | 603.73 | 0.826 |
| **10 mg/L OTC** | 821.74 | 2023.27 | 94.45 | 402.04 | 0.982 |
| **Contribution rate (%) (5 mg/L)** | 21.29 | 57.45 | 8.10 | 13.16 | |
| **Contribution rate (%) (10 mg/L)** | 24.59 | 60.56 | 2.82 | 12.03 | |

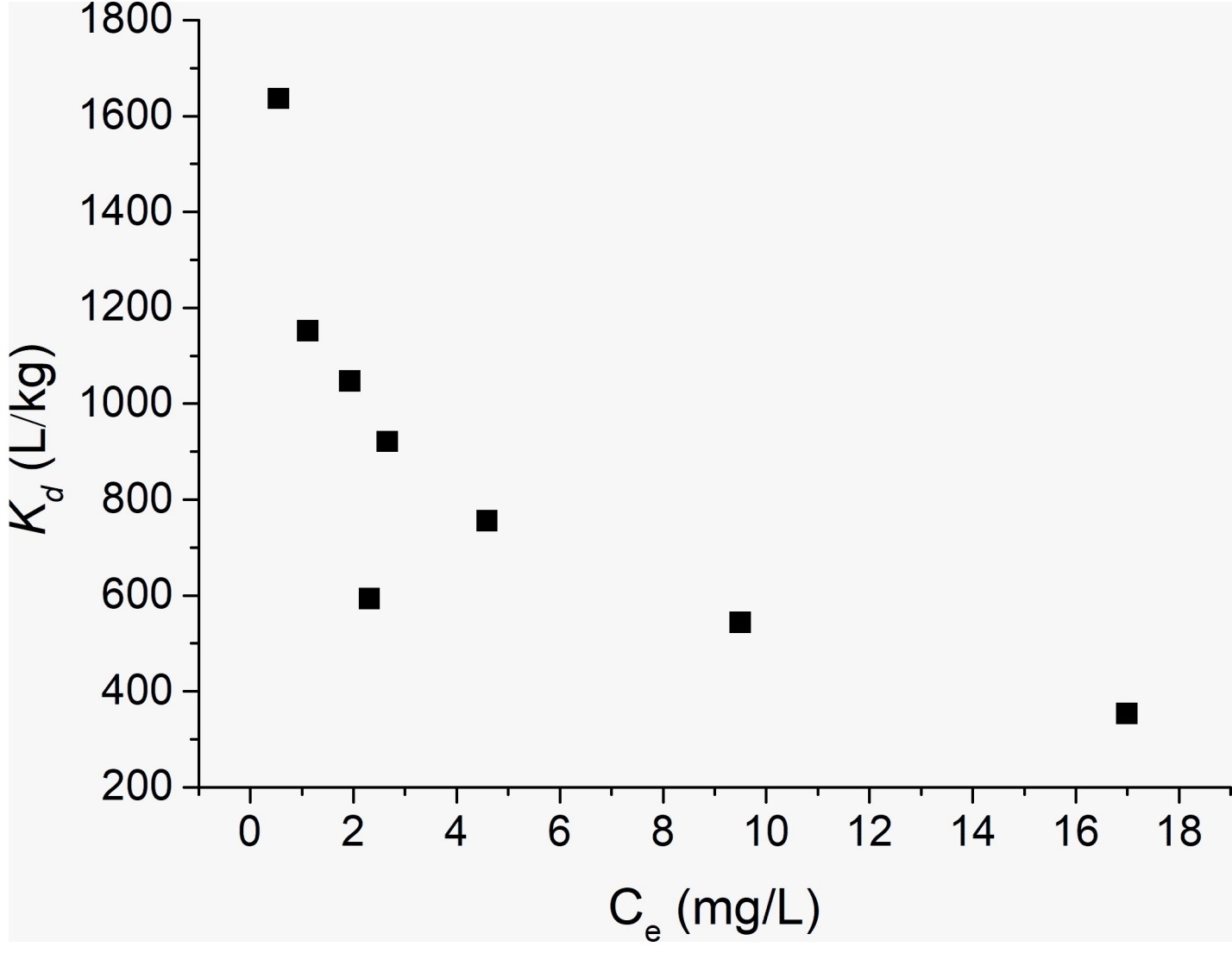

**Fig 7. Values of the adsorption coefficient ($K_d$) for the sorption of OTC on kaolinite.**

## Conclusions

The adsorption of OTC on kaolinite indicated that the adsorption equilibrium was obtained after 24 h, and the adsorption experimental data fit the pseudo-second-order model well. The adsorption isotherms for OTC by kaolinite fit very well with the Langmuir model. The thermodynamic analysis revealed that a spontaneous and endothermic process occurred between OTC and kaolinite. The solution pH had a great effect on the adsorption processes, and a relatively higher adsorption capacity could be obtained for the zwitterionic OTC species. The values of the adsorption coefficient ($K_d$) presented the order of $10^2$–$10^3$.

## Supporting information

**S1 Table. Adsorption kinetics of OTC on kaolinite.**
(XLSX)

**S2 Table. Adsorption isotherms for the OTC adsorption on kaolinite.**
(XLSX)

**S3 Table. Effect of temperature on the sorption coefficient (Kd) for OTC sorption on kaolinite.**
(XLSX)

**S4 Table. Effect of pH on OTC adsorption capacity on kaolinite.**
(XLSX)

**S5 Table. Values of the adsorption coefficient (Kd) for the sorption of OTC on kaolinite.**
(XLSX)

## Author Contributions

**Conceptualization:** Yali Song, Ebenezer Ampofo Sackey.

**Data curation:** He Wang.

**Investigation:** He Wang.

**Supervision:** Hua Wang.

**Writing – original draft:** Yali Song.

**Writing – review & editing:** Ebenezer Ampofo Sackey, Hua Wang.

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
