## [Decision Letter · Decision Letter 0]

9 Sep 2019

PONE-D-19-21989

Adsorption of oxytetracycline on kaolinite

PLOS ONE

Dear Dr song,

Thank you for submitting your manuscript to PLOS ONE. After careful consideration, we feel that it has merit but does not fully meet PLOS ONE’s publication criteria as it currently stands. Therefore, we invite you to submit a revised version of the manuscript that addresses the points raised during the review process.

We would appreciate receiving your revised manuscript by Oct 24 2019 11:59PM. To enhance the reproducibility of your results, we recommend that if applicable you deposit your laboratory protocols in protocols.io, where a protocol can be assigned its own identifier (DOI) such that it can be cited independently in the future. For instructions see: http://journals.plos.org/plosone/s/submission-guidelines#loc-laboratory-protocols

We look forward to receiving your revised manuscript.

Kind regards,

Mohammad Al-Ghouti

Academic Editor

PLOS ONE

Journal Requirements:

* Sorption of tetracycline on biochar derived from rice straw and swine manure, doi: 10.1039/C8RA01454J

In your revision ensure you cite all your sources (including your own works), and quote or rephrase any duplicated text outside the methods section. Further consideration is dependent on these concerns being addressed.

Reviewers' comments:

Reviewer's Responses to Questions

**Comments to the Author**

1. Is the manuscript technically sound, and do the data support the conclusions?

Reviewer #1: Yes

Reviewer #2: No

2. Has the statistical analysis been performed appropriately and rigorously? 

Reviewer #1: Yes

Reviewer #2: No

3. Have the authors made all data underlying the findings in their manuscript fully available?

Reviewer #1: Yes

Reviewer #2: Yes

4. Is the manuscript presented in an intelligible fashion and written in standard English?

Reviewer #1: Yes

Reviewer #2: No

5. Review Comments to the Author

Reviewer #1: This manuscript presents a study about the adsorption of oxytetracycline on kaolinite. It is shown that the adsorption of OTC on kaolinite indicated that the adsorption equilibrium was obtained at 24h and the adsorption experimental data well fitted second -pseudo-order model. The adsorption isotherms for OTC by kaolinite had high fitting with Langmuir model and Freundlich model. The thermodynamic analysis revealed that the adsorption process of OTC on kaolinite was spontaneous and endothermic. The solution pH had a great effect on the adsorption processes and relatively higher adsorption capacity could be obtained for the zwitterionic OTC species. In my opinion it will be of contemporary importance to the readers of PLOS ONE. Hence, upon properly addressing the comments listed below the manuscript should be suitable for publication.

1. On Section 2.3 Data analysis:

The author mentioned that: " The variation of solution pH was from 3.5 to 11.5, and the concentration of OTC was 5mg/L or 10mg/L. ". Since fine kaolinite will change the pH of the solution, it is recommended to specify whether it is the initial or equilibrium pH of the solution.

2. On Section 3.4 The effect of pH on adsorption:

Kaolinite is a 1:1 layer mineral, in which one layer consists of an alumina octahedral sheet and another consists of silica tetrahedral sheet. The charge characteristics of these two layers and edges vary greatly in solutions with different pH values. Therefore, it is suggested to describe in detail the charged characteristics of kaolinite particles at different pH values. It is helpful for the authors to analyze the effect of pH on adsorption capacity for OTC sorption on kaolinite surface.

Reviewer #2: This paper discusses an interesting and relevant topic. Ultimately I would like to see this data published, but I think considerable improvements must be made to the description of methods and analysis before this work is publishable.

The English and language and gramma are poor throughout. This could be improved, this is not a huge job, but it does need to be corrected to improve the clarity and intent of some of the arguments and reduce ambiguity.

Page 8, why SA to 4 decimal places? The SA is very low compared to many kaolinite samples…why? Large average particle size? Very low porosity? How do we know all of the sand and other contaminating minerals were removed? Is there XRD information?

Page 8, what was the pH of the kinetic and isotherm experiments…it would be useful if this was stated in the methods section.

Page 8, is there any replication of the experiments? What is the uncertainty, what controls of data quality were used?

Page 8, what was the ionic strength? The introduction suggests that ionic strength has been tested by other researchers, but there is no mention of the ionic strength for the suspensions in this study.

Page 8, How was the pH controlled for the sorption experiments…adjustments were made between pH 3.5 and 11, but was this measured at the beginning or at the end of the equilibration period? Was there any change in the pH over the equilibration time (it would be unusual if the pH did not change a little)?

Page 8, Why would an intra-particle diffusion model be appropriate here, the SA is very low indicating little porosity, and no estimate of pore volume in given form the BTE SA measurements?

Page 10, I find the results and discussion of a diffusion based mechanism a little puzzling. Given that the porosity is low, and that kaolinite is a classic non-swelling clay, I find it extremely unlikely that there is a diffusion process at play. I do not think there is sufficient evidence to conclude that there is a movement of OTC form external to internal surfaces. Kinetic modelling of this type is really just a curve fitting process, and the models with more adjustable parameters will naturally provide a better fit. Conclusions about actual kinetic mechanisms really need more substantiation that just the fit of a particular semi-empirical model.

Page 11, the commentary on the application of the isotherm modelling is similarly superficial as the kinetic modelling mentioned above, and hence I think there is an over-interpretation of the data sets based on the model parameters, though the overall all conclusion is rather weak in that the authors suggest that there are ‘multiple’ mechanisms responsible for the uptake.

Page 12, the IR spectra and the interpretation is very unconvincing. How were these spectra collected? Is this diffuse reflectance or ATR of dried samples or pastes? There is no description in the methods section, and it is simply not possible to make conclusions from the spectra unless more detail is known about what these spectra represent (ie wet/dry samples, ATR etc). What was the surface concentration of the OTC when absorbed, can you actually see it in the IR, was the OTC just crystalline sample or a solution etc?

Overall I think the commentary through the discussion and analysis of results is too speculative, and this needs significant improvement. I do however, think that much of this could be tidied up quite quickly by the authors thinking more carefully about what the data is telling them. Certainly more description of the methods would help the paper a lot. The data set look very interesting, and is a valuable contribution, though I would insist on some mention of quality control and uncertainty.

6. PLOS authors have the option to publish the peer review history of their article (what does this mean?). If published, this will include your full peer review and any attached files.

Reviewer #1: No

Reviewer #2: No

---

## [Author Response · Author response to Decision Letter 0]

24 Oct 2019

Responds to the editor’s and reviewer’s comments:

Responds to the editor

1. Please ensure that your manuscript meets PLOS ONE's style requirements, including those for file naming. The PLOS ONE style templates can be found athttp://www.journals.plos.org/plosone/s/file?id=wjVg/PLOSOne_formatting_sample_main_body.pdf and http://www.journals.plos.org/plosone/s/file?id=ba62/PLOSOne_formatting_sample_title_authors_affiliations.pdf

Response: We have modified the manuscript according to PLOS ONE’s style requirements.

Response: Considering the editor’s comment, the manuscript has been edited by AJE.

3. We noticed you have some minor occurrence of overlapping text with the following previous publication(s), which needs to be addressed: * Sorption of tetracycline on biochar derived from rice straw and swine manure, doi: 10.1039/C8RA01454J

Response: Some sentences have been modified to avoid overlapping text in the manuscript.

4. PLOS authors have the option to publish the peer review history of their article (what does this mean?). If published, this will include your full peer review and any attached files.

 Response: No

Responds to the reviewer #1

1. On Section 2.3 Data analysis: The author mentioned that: "The variation of solution pH was from 3.5 to 11.5, and the concentration of OTC was 5mg/L or 10mg/L. ". Since fine kaolinite will change the pH of the solution, it is recommended to specify whether it is the initial or equilibrium pH of the solution.

 Response: The solution pH is the initial pH and we have added the description of pH in the manuscript.

2. On Section 3.4 The effect of pH on adsorption: Kaolinite is a 1:1 layer mineral, in which one layer consists of an alumina octahedral sheet and another consists of silica tetrahedral sheet. The charge characteristics of these two layers and edges vary greatly in solutions with different pH values. Therefore, it is suggested to describe in detail the charged characteristics of kaolinite particles at different pH values. It is helpful for the authors to analyze the effect of pH on adsorption capacity for OTC sorption on kaolinite surface.

Response: As the reviewer’s comment, we added the charged characteristics of kaolinite at different pH values inferred from previous studies in the part of discussion. At same time, we discussed how charged characteristics of kaolinite influence the OTC sorption at different pH values.

Responds to the reviewer #2

1. The English and language and grammar are poor throughout. This could be improved, this is not a huge job, but it does need to be corrected to improve the clarity and intent of some of the arguments and reduce ambiguity.

Response: The English language and grammar has been improved. Discussion part has also been tried to clear.

2. Page 8, why SA to 4 decimal places? The SA is very low compared to many kaolinite samples…why? Large average particle size? Very low porosity? How do we know all of the sand and other contaminating minerals were removed? Is there XRD information?

 Response: The equipment of measuring SA gave a 4 decimal value. Considering to small value of SA, the SA is kept 2 decimal places. The kaolinite used in this experiment was bought from Macklin (Shanghai) and pretreatment of kaolinite was relatively simple (just filtered through a 300 mesh sieve), so low SA is obtained. We have found that kaolinite SA of 6.45m2/g was reported by Yandan Li. Most particle size focused on 2.1um and only porosity of kaolinite of 0.0047cm3/g. There is no XRD information for kaolinite sample because of limit of experimental condition.

3. Page 8, what was the pH of the kinetic and isotherm experiments…it would be useful if this was stated in the methods section.

Response: The pH of the kinetic and isotherm experiments is 5.5 and manuscript has been revised for this question. 

4. Page 8, is there any replication of the experiments? What is the uncertainty, what controls of data quality were used?

Response: We have added the contents in section 2.2 in the manuscript.

5. Page 8, what was the ionic strength? The introduction suggests that ionic strength has been tested by other researchers, but there is no mention of the ionic strength for the suspensions in this study.

Response: In this study, 0.01 M CaCl2 was used as background electrolyte of solution and in the manuscript we have added the description of solution background electrolyte.

6. Page 8, How was the pH controlled for the sorption experiments…adjustments were made between pH 3.5 and 11, but was this measured at the beginning or at the end of the equilibration period? Was there any change in the pH over the equilibration time (it would be unusual if the pH did not change a little)?

Response: pH was measured at the beginning of the equilibration experiment. The pH changed by 1 to 2 units of pH after equilibration. 

7. Page 8, Why would an intra-particle diffusion model be appropriate here, the SA is very low indicating little porosity, and no estimate of pore volume in given form the BTE SA measurements?

Response: An intra-particle diffusion model is not appropriate for OTC adsorption of kaolinite, so we have removed the contents about intra-particle diffusion model from the manuscript.

8. Page 10, I find the results and discussion of a diffusion based mechanism a little puzzling. Given that the porosity is low, and that kaolinite is a classic non-swelling clay, I find it extremely unlikely that there is a diffusion process at play. I do not think there is sufficient evidence to conclude that there is a movement of OTC form external to internal surfaces. Kinetic modeling of this type is really just a curve fitting process, and the models with more adjustable parameters will naturally provide a better fit. Conclusions about actual kinetic mechanisms really need more substantiation that just the fit of a particular semi-empirical model.

Response: According to some studies about kinetics of kaolinite adsorption, intra-particle diffusion process is very little. So it is not appropriate to use an intra-particle diffusion model here and the contents have been deleted from the manuscript.

9. Page 11, the commentary on the application of the isotherm modeling is similarly superficial as the kinetic modeling mentioned above, and hence I think there is an over-interpretation of the data sets based on the model parameters, though the overall all conclusion is rather weak in that the authors suggest that there are ‘multiple’ mechanisms responsible for the uptake.

Response: We have revised some detail description of isotherm model and discussion has been added.

10. Page 12, the IR spectra and the interpretation is very unconvincing. How were these spectra collected? Is this diffuse reflectance or ATR of dried samples or pastes? There is no description in the methods section, and it is simply not possible to make conclusions from the spectra unless more detail is known about what these spectra represent (ie wet/dry samples, ATR etc). What was the surface concentration of the OTC when absorbed, can you actually see it in the IR, was the OTC just crystalline sample or a solution etc?

Response: We have added the method of IR spectra collection in section 2.1. In this study, KBr pressed-disc method was adopted. OTC of 10mg/L adsorbed by kaolinite was used and all samples for IR measuring were solid. 

References

1. Yandan Li. Sorption behavior of typical flueroquinolone antibiotics on kaolinite：batch experiments. Bioresour Technol. Chia university of geosciences. 2017.

---

## [Editor Report · Decision Letter 1]

4 Nov 2019

Adsorption of oxytetracycline on kaolinite

PONE-D-19-21989R1

Dear Dr. song,

We are pleased to inform you that your manuscript has been judged scientifically suitable for publication and will be formally accepted for publication once it complies with all outstanding technical requirements.

With kind regards,

Mohammad Al-Ghouti

Academic Editor

PLOS ONE
---

## [Editor Report · Acceptance letter]

8 Nov 2019

PONE-D-19-21989R1 

Adsorption of oxytetracycline on kaolinite 

Dear Dr. Song:

I am pleased to inform you that your manuscript has been deemed suitable for publication in PLOS ONE. Congratulations! Your manuscript is now with our production department. 

With kind regards,

on behalf of

Dr. Mohammad Al-Ghouti 

Academic Editor

PLOS ONE